# AO-Net: Efficient Neural Network for Ambient Occlusion

Jiayi Wang*
Tsinghua University

Fan Zhou†
Tsinghua University

Xiang Zhou‡
Tsinghua University

Yongle Li§
Tsinghua University

Xueqian Wang¶
Tsinghua University

## ABSTRACT

Screen space based ambient occlusion is widely applied in real-time 3D applications due to its high efficiency, however, it frequently exhibits artifacts including banding and blurring. In this paper, we propose AO-Net, a learning-based method for fast and high-quality ambient occlusion generation. Our neural network is built upon kernel prediction-based architecture with careful input screen space feature selection, leading to a light-weight and compact solution. Experiment results indicate that our approach can achieve visual quality to a level in comparable with ray-traced solutions, meanwhile maintaining real-time performance. In addition, our method can be easily integrated into existing rendering pipelines and shows robustness for unseen scenes.

**Index Terms:** Ambient occlusion—Neural networks—Global illumination—Shading;

## 1 INTRODUCTION

Ambient occlusion(AO) plays a crucial role in realistic rendering applications and is a widely-adopted approximation to global illumination. It can greatly enhance viewers' 3D perception by darkening locations via adjusting obscurance coefficients [1]. As illustrated in Fig. 1, the AO value of position $p$ with normal $n$ can be calculated by integrating the visibility function over the hemisphere $\Omega$. One key observation is that the AO values are view independent and only relate to scene geometry.

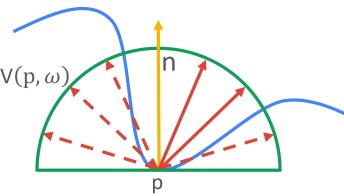

Figure 1: Mathematical definition of AO. The AO value of $p$ with normal $n$ equals the integration of visibility function $v(p,m)$ over the upper hemisphere. The visibility of dashed arrows equals to 0, otherwise 1.

High-quality AO can be obtained via hardware-assisted ray-traced techniques [2], which are usually time-consuming and difficult to use in real-time applications due to the limited power budget. In order to achieve high performance, screen-based solutions that solely rely on depth or/and normal buffer as input are widely adopted. To name a few, Screen Space Directional Occlusion(SSDO) [3], Ground Truth-based Ambient Occlusion(GTAO) [4] and Horizon-based Ambient Occlusion (HBAO) [5] are all approaches of this kind. These approximations, however, often lead to low visual quality due

*e-mail: jiayi-wa21@mails.tsinghua.edu.cn

†e-mail: zhouf21@mails.tsinghua.edu.cn

‡e-mail: zhoux21@mails.tsinghua.edu.cn

§e-mail: liyl21@mails.tsinghua.edu.cn

¶e-mail: wang.xq@sz.tsinghua.edu.cn

to the absence of global information and are prone to exhibiting artifacts such as blurring and banding.

In recent years, important advances incorporating neural networks into computer graphics have been made, especially in tasks such as Monte Carlo imagery denoising [6, 7], super sampling [8], and light field [9]. Learning-based approaches [10, 11] have shown potential to produce AO. However, both quality and efficiency need to be improved to meet the demands of real-time rendering. In addition, the generalization and robustness of deep learning-based approaches have not been carefully studied.

In our work, we propose a novel method for AO which aims at obtaining ray-traced quality and maintaining real-time performance. Our framework generates AO by a deep learning network from screen space buffers, which can be easily obtained in a deferred rendering pipeline. First, we investigate the sensitivity of screen space buffers with respect to the quality of AO and exclude redundant features that have little influence to keep the memory bandwidth as small as possible. Second, the artifacts, for instance, blurring and banding can be addressed via a kernel-prediction neural network. In the training phase, we use ray-traced AO as targets. In addition, we build up a prototype by integrating our trained model into a rasterization real-time rendering pipeline. We validate our approach on various seen and unseen scenes.

To summarize, our work has made the following contributions:

- **A lightweight and compact network** built upon kernel prediction-based architecture, which eliminates blurring and banding artifacts, and achieves ray-traced visual quality at real-time frame rates.

- **A careful input screen space features selection** for AO tasks, which reduces redundancy and improves the network efficiency.

- **An end-to-end AO solution** that produces high-quality AO results by combining a deferred rendering pipeline.

## 2 RELATED WORKS

Ray-traced Ambient Occlusion   Early methods used ray-tracing to generate AO [12–14] for static scenes, which more accurately accounted for ambient light. These methods are used as preprocessing steps in real-time interactive applications, but are limited to static scenes and are slow to compute. In order to support dynamic scenes, Kontkanen [15] et al. proposed a precomputed AO field for rapid calculation of AO, which is further applied in approximating the AO of dynamic objects. Bunnell [16] proposed a GPU-based deformed surface method to deal with dynamic cases. Christensen [17] extended this approach and used it to calculate diffuse global illumination. All these methods are based on various surface discretization or ray-tracing. For AO method based on ray-tracing, it is of utmost importance to increase calculation speed.

Screen Space Ambient Occlusion   Screen Space Ambient Occlusion (SSAO) method was first introduced by Mittring [18] to avoid the huge amount of calculation and additional storage overhead of ray-traced AO methods. By sampling from the depth buffer in the view sphere space, SSAO calculates the number of occlusion points within the depth surface, and estimates the occlusion factor. It's widely used in modern 3D applications, but due to the large

difference in depth of neighboring pixels in screen space, it sometimes causes artifacts. Bavoi et al. [5] introduced Horizon Based Ambient Occlusion (HBAO). The rays travel along the depth buffer, and HBAO is calculated based on the depth difference. However, this method only considers the visible points in the current tracking perspective, which will result in incorrect or missing shadows. An important strategy was adopted by Vardis et al. [19] to identify key areas of sampling through multi-view joint optimization. Jimenez et al. [4] propose GTAO, to speed up calculations and improve accuracy through a new form of integration. A common problem with the screen space AO methods described above is that the thresholds or experience sampling parameters must be carefully chosen to avoid artifacts.

**Deep Learning for Ambient Occlusion** In recent years, with the successful application of neural network, it is used to solve rendering problems. To solve the AO problem, Holden et al. [20] first introduced a neural network called NNAO that uses a neural network to predict an approximation of Ambient occlusion. NNAO network is merely a 4-layer MLP, so the learning ability of NNAO is poor, and high-quality AO results cannot be produced. Nalbach et al. [10] introduced Deepshading, a convolutional neural network, which learns the mapping from the view-space position and normal to ray-traced AO image. However, this method has low efficiency and poor generalization performance. Zhang et al. [11] proposed DeepAO, which reduces the number of network layers and parameters based on U-Net networks, further improving rendering efficiency. But it sacrifices rendering quality to a large extent so the rendering quality still needs to be improved.

**Kernel Prediction** Kernel Prediction has demonstrated significant effectiveness in the field of denoising in Monte Carlo rendering to remove noise such as indirect lighting, soft shadows, etc. Bako et al. [21] used convolutional neural networks to predict the filter kernel and yielded more significant results in denoising effects. Vogels et al. [7] use multiple deep residual networks to predict multiple filter kernels at different scales. We first apply kernel prediction module to the field of AO generation, where the features extracted by U-Net network are used to generate kernel weights for AO. We also compare our method with original U-Net, which confirms that kernel prediction in AO generation tasks is essential for removing noise from sparsely sampled images, leading to further improvement in rendering effects.

## 3  OUR METHODS

Our approach is illustrated in Fig.4: given G-buffers from a deferred rendering pipeline at a certain view, we seek to generate its AO pass. Consequently, we first describe the screen space feature selection in Sec.3.1 via a sensitivity study, and then discuss the network architecture in Sec.3.2 and Sec.3.3. Finally, we discuss the results on various scenes in Sec.4.2 before concluding in Sec.5.

### 3.1  Feature Selection

We use supervised learning to train our neural network. The training pairs consist of features generated by rasterization and reference AO generated by ray-tracing. Previous learning-based AO generation methods [10, 11] used view-space normal and depth in G-buffer as input to the network, which is usually time-consuming for network inference and thus difficult for real-time applications. As mentioned in Sec.1, the AO values only relate to the neighboring geometry information(normal and depth). On the one hand, it is also interesting to note that normal and depth are indeed correlated: normal is the cross product of gradients of the depth buffer in smooth regions. On the other hand, the visible view-space normal always points to the camera. So we believe that the used features can be further reduced.

Our feature selection method is inspired by Neural Shadow Mapping [22], which uses sensitivity analysis to simplify the neural

| Feature | Relative Sensitivity |
|---------|---------------------|
| x | 0.0204 |
| y | 0.0328 |
| z | 0.0308 |
| depth | 0.0089 |

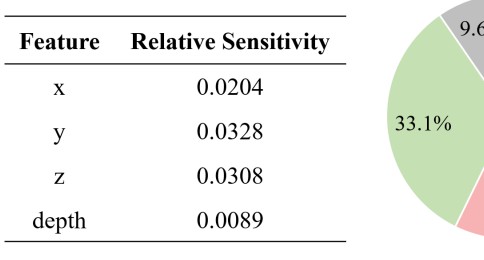

Figure 2: Sensitivity Analysis. The relative sensitivity of the input features, including depth and normal which consists of three dimensional x, y and z. We can observe that normal plays a more important role in prediction.

network input by quantifying the importance of each input channel. Sensitivity analysis is used to evaluate changes in network output caused by small perturbations in the input channel. Intuitively, if the channel contributes more to interpreting changes in output, the sensitivity is higher. The absolute sensitivity $S_i$ of the input channel $x_i$ is expressed by

$$S_i = E\left[\frac{\phi(x_i + \xi_i) - \phi(x_i)}{0.1\sigma_i}\right], \xi_i \sim N(0, 0.1\sigma_i), \quad (1)$$

where $\phi$ is our network, and $\xi_i$ is randomly generated Gaussian noise with the mean value of 0 and the standard deviation of $\sigma_i$, which adds random perturbations to the $i^{th}$ channel. $\sigma_i$ is the standard deviation corresponding to the $i^{th}$ channel, which is estimated empirically by aggregating all pixels in the dataset of each channel. Finally, we calculate the relative sensitivity $s_i = S_i / \Sigma_i S_i$ to compare the sensitivity of the different input channels.

We systematically evaluate and select all channels including depth and three dimensions of normal by calculating the sensitivity of each channel, as shown in Fig. 2. In order to ensure the consistency of the perturbation, we select normalized depth for the experiment, so that the scale of depth is the same as the scale of normal. It shows that the perturbation of normal has a great influence on the output results, indicating channels of normal play a vital role in the network. In contrast, the depth channel has little effect on network results, less than 10%. Subsequent ablation experiments demonstrated the effectiveness of our feature selection, as shown in Table 3. Moreover, depth varies greatly in different scenes, making it difficult for the network to converge. Therefore, we choose normal as the input to

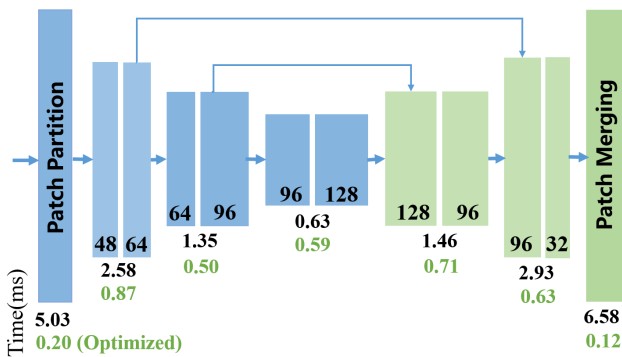

Figure 3: Layer-wise performance optimization before and after for a 720×1280$px$ input.

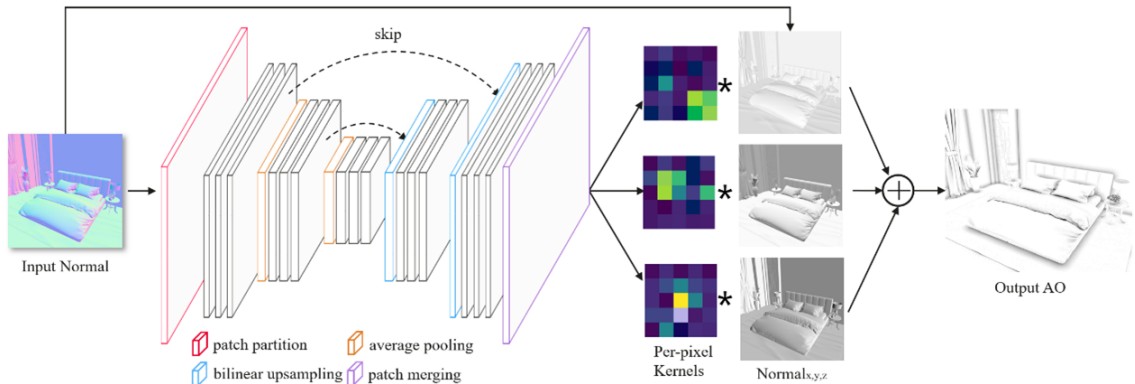

patch partition    average pooling

bilinear upsampling    patch merging

Figure 4: Overview. Given a normal map, we first partition it into multiple patches and input them into U-net to generate kernel weights. We then use the predicted kernel weight to locally weigh the input normal, and obtain the output AO.

our network while the output is compared against ray-traced AO as the target during training.

## 3.2 Patch-based U-Net

Vanilla U-Net yields a network with low performance (>20ms), which can not be adapted to the real-time graphics pipeline. Therefore, the primary goal of our approach is to optimize the network structure to improve efficiency with minimal impact on quality.

We first analyze the performance of each layer of the network. Fig. 3 shows the time spent at each layer of the network, which takes a total of 20.56ms. We see that the first layer (including encoder-decoder) takes 11.61ms, which takes the longest time and accounts for more than 50% of the time for the entire network. In addition, we observe that as we move from the inner layer of U-Net to the outer layer, the number of channels is reduced by half and the resolution is doubled. In practice, as the resolution increases, the run-time of the layer increases significantly. Therefore, we believe that the key solution to reducing network run-time is to optimize the first layer, which is the most time-consuming. In addition, reducing the resolution of each layer of the network is critical to improving network performance. A common approach is to replace the first layer with a simple downsampler and upsampler in the U-Net. However, it can cause some crucial information be lost in the input and can produce less clear output or even artifacts.

Therefore, a patch-based network is proposed to optimize network performance. We optimize the first layer (including encoder-decoder) which is the most time-consuming. On the encoder side, we use a patch partition to flatten each input 4x4 square of pixels as

a patch to 16 separate channels and send the (h/4 × w/4) patches directly to the second layer. Therefore, the input buffer size is converted from (h × w × c) to (h/4 × w/4 × 16c). We do the opposite on the decoder side: rearranging the 16 output channels to 4x4 pixels through patch merging. Patch-based networks replace the most time-consuming first layer, which greatly improves network speed. After our optimization, the run-time of the first layer network is reduced from 11.64 ms to 0.32 ms. The patch-based method optimizes the network performance to 3.60 ms, which is only 18% of the initial time.

Fig. 5 shows the speed and quality comparison before and after network optimization. It shows that our network has a significant increase in network speed with minimal impact on quality. We optimized the network structure so that the network has high performance (<4ms, resolution: 720 ×1280) and requires little memory (<1.5MBs), allowing the network to be easily integrated into the rendering pipeline. However, the large compression of time also leads to a slight loss of quality, reducing the network's ability to preserve details, so we incorporate a kernel prediction module to improve the quality and detail of the generated AO.

## 3.3 Kernel Prediction Module

Furtherly, we incorporate the kernel prediction module, which has demonstrated significant effectiveness in the field of denoising [7, 21], to alleviate the above-mentioned problems and enable our AO-Net generating fine-level details.

Kernel prediction uses neural networks to estimate the locally weighted kernel of each pixel from neighboring pixels. By concatenating U-Net in series with kernel prediction module, the features extracted by our patch-based U-Net are directly used to generate weighted kernels. We use the predicted kernel weight to locally weigh the input normal, and obtain the output AO.

The output of kernel prediction has $K^2N$ channels, $K$ is the size of linear filters at each pixel and $N$ is the number of input channels per pixel. The value $Y^p$ at each pixel p can be expressed as

$$\hat{Y}^p = \frac{1}{N} \sum_{i=1}^{N} \left\langle f_i^p, V^p(X_i) \right\rangle, \tag{2}$$

where $V^p(X_i)$ is the $K \times K$ neighborhood of pixel $p$ in image $X$ of the $i$-channel, and $f$ is its corresponding kernel. $<,>$ means a dot product then calculate average.

Our approach combines the advantages of U-Net and kernel prediction. It preserves more local details meanwhile achieving a large receptive field, resulting in a output with sharper contour. In addition, it ensures that the final output estimate always lies within a

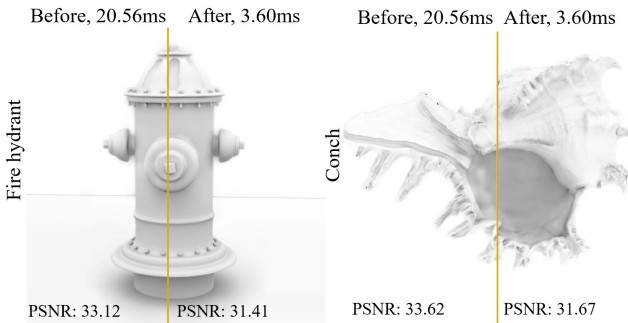

Before, 20.56ms   After, 3.60ms     Before, 20.56ms  After, 3.60ms

Fire hydrant    Conch

PSNR: 33.12    PSNR: 31.41     PSNR: 33.62    PSNR: 31.67

Figure 5: Figure demonstrating the effect of our proposed optimizations. Our patch-based network achieves a significant increase in network speed with minimal impact on quality.

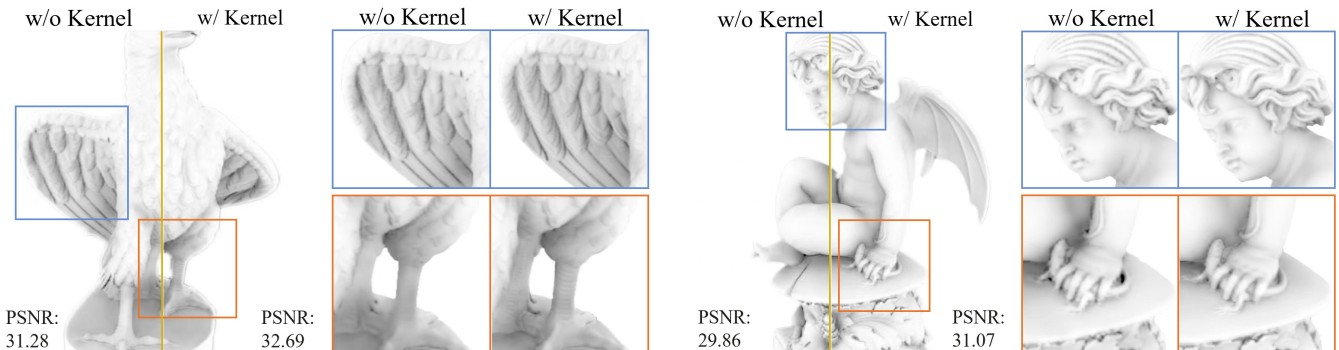

Figure 6: Ablation study on kernel prediction. It can be observed that kernel prediction preserves more detail and contrast and produces sharper results.

convex hull of the respective neighborhoods of the input image. This greatly reduces the search space of the output value, thus avoiding the artifacts generated by U-Net direct prediction. (e.g., color shifts). Similar to Bako et al. [21], each filter uses the decoder block of U-Net to predict a 4 × 4 kernel that is applied to the input image.

We compare our kernel prediction based method with U-Net direct prediction which is shown in Fig. 6. The results show that the AO directly predicted by U-Net retains the blurring and artifacts. After the introduction of kernel prediction, the network preserves more local details, which further improves the rendering quality.

### 3.4 Loss Function

In this section, we detail loss functions that are employed in the training process as below.

**Content loss**   In previous AO generation methods, SSIM [23] loss function is commonly used to minimize the error between the predicted AO and the ground truth. Although large SSIM weights can speed up the rate of convergence, they also cause excessive smoothness of rendering results. To avoid this, we set 0.5 as its weight:

$$l_{DSSIM} = 0.5(1 - l_{SSIM}). \qquad (3)$$

**Perceptual loss**   Content loss compares differences in spatial similarity, but it is less sensitive to detail. We use perceptual loss in Equation 4 to produce sharper edges and improve the image detail of the rendering result. VGG-19 [24] perceptual loss function can minimize the distance between the actual output and generated value in the feature space, so we use the first several layers feature map of VGG-19 network to obtain better visual quality. The perceptual loss is defined as

$$l_{per} = ||\Phi VGG(y) - \Phi VGG(G(x))||_2, \qquad (4)$$

where $\Phi$ denotes activation of the $i$-th layer in the VGG-19 network.

In summary, the overall loss of AO-Net is defined as

$$L_G = \lambda_1 l_{per} + \lambda_2 l_{DSSIM}. \qquad (5)$$

## 4 EXPERIMENT

### 4.1 Experimental Settings

**Datasets**   a) Our dataset: consists of 3996 pairs of deferred shading G-buffer data and corresponding reference images in a resolution of 720 × 1280 px. To create the dataset, 37 scenes are utilized to generate the pair annotated data, the scenes include food, animal, furniture, building, etc. In these 3996 pairs, we use 3200 pairs to train the network, 398 for validation, and the remaining 398 for testing. The training and validation images share the same set of 29 scenes, while the test images come from 8 different scenes

not used for training or validation. b) DeepAO dataset: provided by Zhang et al. [11], which includes 17 scenes and 42,000 pairs of deferred shading G-buffer data, with corresponding reference AO results. 32,000 pairs of images are used to train the network while 5,000 pairs are used for validation, and the remaining are used for testing. The training and validation images share the same set of 14 scenes, while the test images come from the remaining 3 scenes. The training and test scenes are disjoint.

**Baselines**   Using ray-traced AO as a reference, we compare the rendering effect and runtimes with the current state-of-the-art methods, which proves the effectiveness of AO-Net. We compare against a) state-of-the-art neural network AO methods: Deepshading [10] and DeepAO [11]. b) classic screen space AO methods: GTAO [4] generated with eevee in Blender.

**Implementation details**   We use Pytorch framework in implementation and employ one NVIDIA Tesla v100 GPU for both training and inference. We train our model using Adam optimizer with an initial learning rate of 1e-3. During U-Net upsampling, we use bi-linear interpolation instead of expensive transpose convolutions to further speed up inference. In addition, we use Average-pool instead of Max-pool to improve temporal stability. The loss weights in Equation 5 are set to $\lambda_1 = 0.06$, $\lambda_2 = 0.94$ . The mini-batch size for all scenes is set at 16 in training.

### 4.2 Result

**Quantitative Comparisons**   The statistical results of our experiment are shown in Table 1. We use SSIM and PSNR to evaluate the rendering quality and mark the best results in bold in the table. "-" means that the running time of GTAO has not been measured, because it is difficult to measure the running time of this process in Blender. It can be seen that our method outperforms other comparison methods on quality, and is superior to all state-of-the-art neural network AO methods on speed.

**Qualitative Comparisons**   We randomly selected several scenes to compare the visualization results of various algorithms, as shown in Fig. 7. We can observe that our method is superior to other advanced algorithms in overall visual effect and predicts AO results

Table 1: A quantitative comparison of our model with three SSAO-based methods on the datasets: DeepAO [11] and our new dataset.

| METHODS | DEEPAO DATASET | | OUR DATASET | | Run-time(ms) |
|---|---|---|---|---|---|
| | SSIM↑ | PSNR↑ | SSIM↑ | PSNR↑ | |
| DeepAO | 0.8794 | 21.75 | 0.9470 | 24.40 | 3.98 |
| Deepshading | 0.8892 | 23.55 | 0.9597 | 25.71 | 6.09 |
| GTAO | 0.9058 | 23.79 | 0.9767 | 26.10 | - |
| OURS | **0.9223** | **25.10** | **0.9802** | **32.59** | **3.92** |

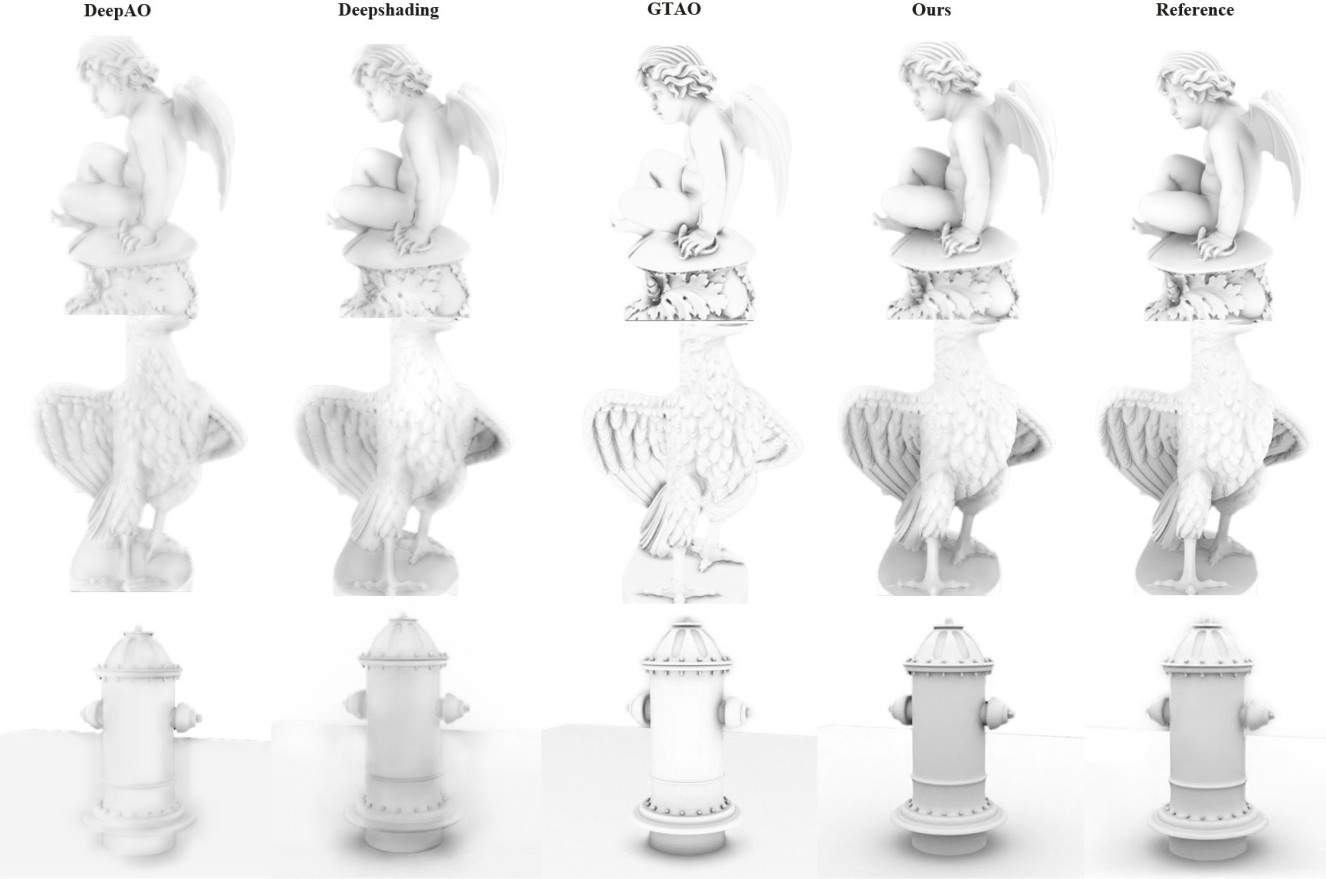

|        |        |       |      |           |
| DeepAO | Deepshading | GTAO | Ours | Reference |

Figure 7: Comparison with other SSAO-based methods. The experimental results show that our method outperforms other advanced methods in overall visual effect and predicts AO results closest to the ray-traced level.

closest to the ray-traced level. In addition, our method performs best in occlusion details along object edges and at corners, as shown in Fig. 8. We demonstrate that our approach can lead to more accurate generation and better image detail.

### 4.2.1 Run times at different resolutions

In addition, we tested the average run time at different resolutions. We mark the best performance in Table 2 in bold. It can be seen that our method is superior to other methods on speed, especially at high-resolution input.

Table 2: Timing in ms for three learning-based AO methods on different resolution.

| Method | $540 \times 540px$ | $720 \times 1280px$ | $1024 \times 2048px$ |
|---|---|---|---|
| DeepAO | **2.26** | 3.96 | 7.09 |
| Deepashading | 3.56 | 6.09 | 11.35 |
| OURS | 2.67 | **3.92** | **6.95** |

### 4.2.2 Ablation Study

We first analyze the validity of our feature selection. Next, we ablate the impact of patch-based optimization and kernel predict module on reference time and quality.

**Ablation study on input features**   To validate the effectiveness of our feature selection, we compare three different kinds of input features and report metrics averaged over our dataset in Table 3.

We observed that with normal input alone we can get high-quality AO while the addition of depth has little impact on the quality of the rendered image. As mentioned in Sec.3.1, normal is the cross product of the gradient of the depth buffer in the smooth region, so normal can be seen as knowledge distilled from depth, and is sufficient to express the geometric information required for AO generation. More importantly, because the depth of different scenes varies greatly, the quality can degrade greatly if depth is not normalized or is not normalized correctly, thus we consider the instability of depth information to play a negative role in AO generation tasks. In addition, only normal input improves the network efficiency and accelerates the network convergence speed in the experiment.

Table 3: Ablation study on input features. Only normal input achieves the highest accuracy and shortest run time. In addition, unnormalized depth may have a negative impact on the accuracy of the network.

| Input Features | SSIM↑ | PSNR↑ | Run-time(ms) |
|---|---|---|---|
| Only Normal | 0.9802 | 32.59 | 3.92 |
| Normal+Depth | 0.9743 | 30.51 | 4.22 |
| Normal+Depth(Normalize) | 0.9786 | 32.45 | 4.22 |

**Ablation study on Network Architecture**   We designed experiments to verify the validity of our proposed patch-based optimization scheme and kernel prediction module. As shown in Table 4, *Base* works as a baseline which use original U-Net to predict rendering results. We train the model *Base+Patch* that includes the patch-

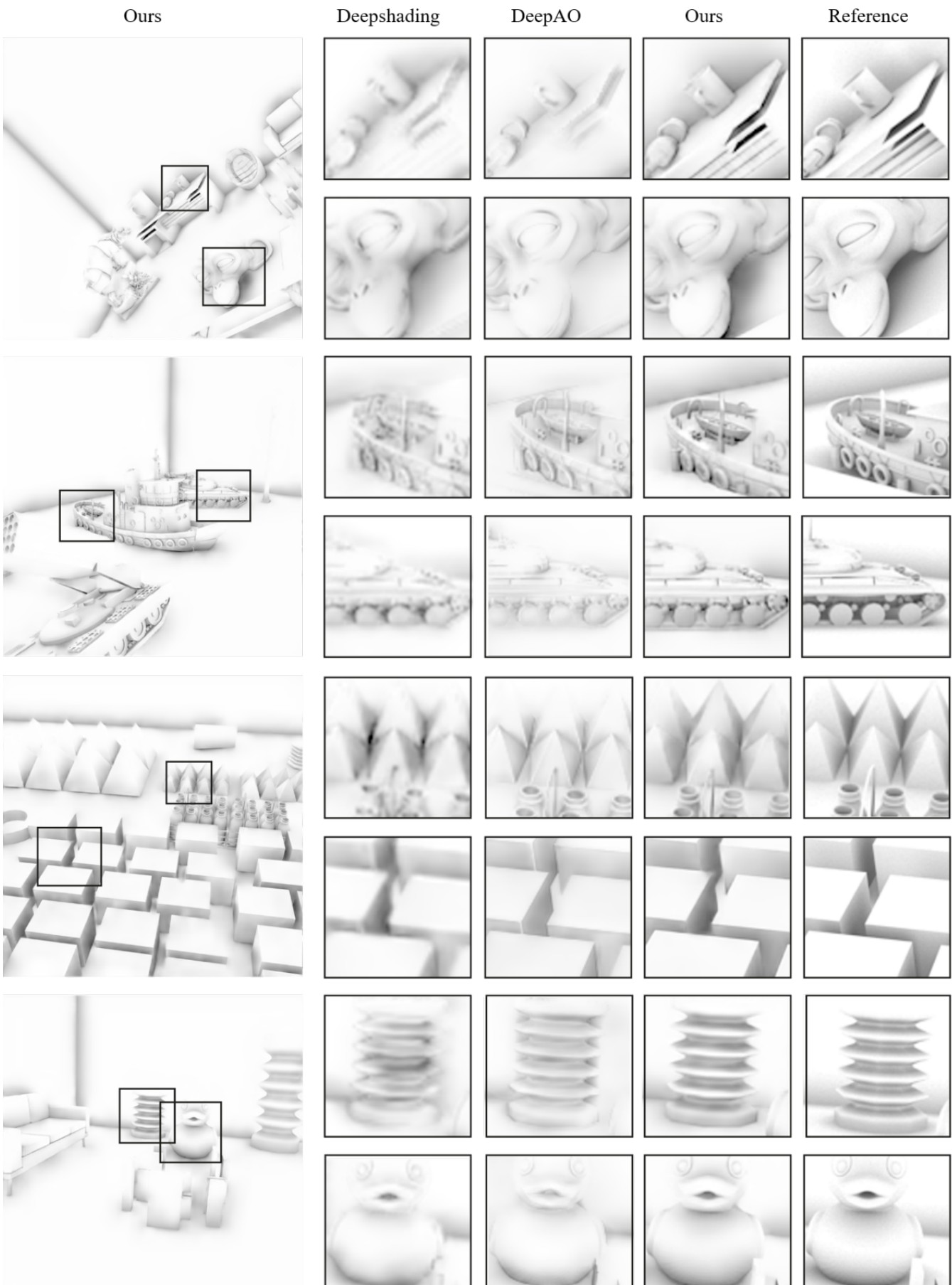

Figure 8: Visual comparison of different learning-based methods.

based optimization above the model *Base*. Compared with *Base*, *Base+Patch* reduce the time consumption to 18%, only 3.60ms. In addition, kernel prediction module can further optimize the rendering quality without incurring a greater loss of efficiency. More importantly, it avoids blurring and artifacts from direct predictions, as shown in Fig. 6. *Base+Kernel+patch* combines kernel prediction with patch-based optimization to generate high-quality renderings results at a fast speed and achieve optimal results.

Table 4: Ablation study on patch-based optimization and kernel prediction. We can observe that: 1) Patch-based optimization method can achieve the highest efficiency. 2) Kernel prediction module achieves the highest accuracy 3) The combination of both achieves a balance between speed and quality.

| Models | SSIM↑ | PSNR↑ | Run-time(ms) |
|---|---|---|---|
| Base(U-Net) | 0.9880 | 33.40 | 20.55 |
| Base+Patch | 0.9732 | 30.94 | 3.60 |
| Base+Kernel | 0.9887 | 35.50 | 22.78 |
| Base+Patch+Kernel | 0.9802 | 32.59 | 3.92 |

## 5 CONCLUSION

We present a learning-based method for AO generation. The compact input of our solution is designed via an investigation of different screen space information's influences on the AO quality. Furthermore, the use of kernel prediction-based architecture further improves the visual quality to ray-traced level. We demonstrate that our method can reach real-time frame rates by integrating with a deferred rendering pipeline. Various examples indicate that our methods are robust to unseen scenes.

However, our work has the same limitations as all screen space solutions. G-buffer does not contain complete information about the scene, which may lead to defects on screen boundaries or special perspectives. In the future, we would like to explore using global information to further improve the quality and other scenarios such as stereo rendering.

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
