# OpenReview forum: "AO-Net: Efficient Neural Network for Ambient Occlusion"
_graphicsinterface.org/Graphics_Interface/2023/Conference — GI 2023_

### Official Review · Reviewer_HCFE · 2023-01-03
**This paper proposes a lightweight and efficient method of generating ambient occlusion (AO) maps in screen space. The method is able to achieve better result compared with existing DL AO methods, such as DeepAO, while maintaining realtime framerates.**

**Rating:** 7
**Confidence:** 4

**Review:**

This paper demonstrates a lightweight and efficient method of generating ambient occlusion (AO) maps in screen space from the depth buffer. Utilizing the neural network, multiplexing feature selection and kernel prediction, the method can generate high-fidelity AO maps close to the reference ray-traced results, with low time cost. The authors examine the time efficiency and performance by comparing them to similar AO map generating methods.

This novelty of the paper is however only marginal. The feature selection and kernel prediction modules mainly behave as enhancing the result of an updated U-Net, originating from DeepAO [1]. 2. The expression of this paper is very rough and lacks detailed illustrations. For example, the illustration of dataset generation used in the examination is too simple. The time performance is very close to DeepAO, whereas the storage consumption may be higher. A discussion of memory runtime consumption is missing here.

Overall I think this is a good candidate for GI 2023 and recommend acceptance.

---

### Official Review · Reviewer_benL · 2023-01-09
**A paper with a weak novelty but a practical neural network for a screen-space ambient occlusion**

**Rating:** 6
**Confidence:** 4

**Review:**

The paper presents an efficient neural network that infers a screen-space ambient occlusion map from shading normals. It follows typical supervised learning and lets the neural network learn a mapping from computationally cheap information (i.e., normals) to a reference (i.e., ray-traced ambient occlusion).

The proposed framework is relatively straightforward and not quite novel since the framework consists of well-known mechanisms such as U-Net and a kernel-predicting layer without noticeable adaptations. The limited novelty of the paper is the primary concern.

The main strength of the paper is its practical value. The presented results are convincing, with a real-time inference time (e.g., ~4ms for 1280 x 720 images). Also, the framework is quite simple, and I think it is straightforward to implement it.

I want to suggest the following modifications to the paper.

The authors fed only normals to the neural network based on an empirical study where normals are more critical than depths for predicting screen-space ambient occlusion maps. A more straightforward alternative is to feed both normals and depths to the network and train the neural network to extract comprehensive features from both features. One of the ablation studies, Table 3, suggests that including the depth as an additional input to the U-Net is useless. It needs more technical discussion as this result is not intuitive. Some image parts can benefit from the depth, especially where normals do not faithfully guide the ambient occlusion. I would like to see a technical discussion that further motivates the design choice of the network input (i.e., only normals).

Sec. 3.2 discusses the computational overhead of the U-Net and motivates an optimization based on the overhead. However, the computational overhead of the network inference varies according to the exact configuration of the chosen U-Net. Knowing the parameter details of the U-Net (Fig. 3) is necessary to agree with the current motivation.

Overall, the presented ideas (e.g., a slight modification of the first layer in the U-Net and incorporating a kernel-predicting layer into the network) are relatively straightforward. Nevertheless, the presented network is quite simple and introduces a practical gain over the previously tested methods.

---

### Official Review · Reviewer_pshb · 2023-01-13
**solid advance**

**Rating:** 7
**Confidence:** 3

**Review:**

This paper presents a dedicated neural architecture for ambient occlusion. Initially based on U-net, the network uses a patch-based layer to speed up the initial high-resolution layer, and includes a kernel prediction module to enhance details. An ablation study shows that the patch module reduces runtime considerably at only a minor reduction in quality. PSNR and SSIM comparisons with recent neural approaches to AO show a quality advantage to the proposed approach.

The paper presents a plausible approach to network design and shows good performance improvement over the chosen comparators. It is reasonably organized. Overall, I recommend that the paper be accepted.

The authors are careful to mention that the training and testing scenes in their dataset are disjoint. However, there is no discussion of reserving scenes for testing in the much larger DeepAO dataset (5k image pairs are reserved for testing, but no mention is made of which scenes they come from). Data leakage is always a concern for neural approaches; if the same scenes were used in training and testing, that could account for part of the results. If the training data used the same scenes, this is a significant problem with the protocol and may have exaggerated the effectiveness of the method; in such a case, I would recommend that the authors rerun the study while reserving some scenes strictly for testing.

The authors report quite different sensitivity to the x and y components of the normal vector. This is unintuitive. Can the authors comment on this? It may be an artifact of the scenes used, in which case greater generalization could be achieved by either using greater diversity of scenes or training with rotated images.

How were the PSNR and SSIM figures computed? A single number is given to summarize a huge number of data points. It could be interesting to see some commentary about the distribution. Are there scenes where the method does not work as well? If so, that could point to future work.

It is rather harder than necessary to evaluate what is new in this paper and what was taken from previous work. The authors use the word "introduce" a few times; the usual meaning is that the thing introduced is novel (invented by the authors) but that is not the usage here. Consider revising these statements.

Minor comments:

Do not use "etc" in the abstract. Provide a complete list, if possible.

3.1: "In conclusion, we believe the used features can be further reduced." Why "in conclusion"? This claim does not seem to follow from the preceding discussion.

Equation 3 uses a literal "0.5". Given that there is already a parameter lambda_2 weighting this term, is there any point to the extra constant?

In Table 1, the runtime of GTAO is given as "-". What does this mean? The obvious meaning is that the method takes zero time, which is probably not correct.

You don't report times needed for training, as far as I can tell. The training regime is described in 4.1, but no summary figure is given. Just one sentence would be enough.

Try to fix up the bibliography. The capitalization is sometimes incorrect ("Gpu", "monte carlo"). SIGGRAPH should be all caps. The entries switch between using initials and spelling names in full. References 4 and 24 are incomplete. In general, there are many small errors that could be corrected.

The paper would benefit from an editing pass for typos and grammar.

---

### Meta-Review · Area_Chair_S3Eh · 2023-01-15

**Recommendation:** accept
**Confidence:** 5

**Metareview:**

A neural architecture is proposed for real-time ambient occlusion. The network uses a patch-based layer to speed up the initial high-resolution layer, and includes a kernel prediction module to enhance details. The strengh of the paper lies in the simplicity and practical value, and the weakness is the limited novelty.  Overall the reviewers are positive. We encourage the authors to include the following changes in the revision.
1. rerun the study while reserving some scenes strictly for testing;
2. add technical discussions on the results of Table 3 that suggest including the depth as an additional input to the U-Net is useless;
3. try your best to respond to other issues raised in the reviews.